# On the Relationship between a Student Association's Endeavors to Foster a Social–Academic Climate on Campus, Students' Self-Efficacy, and Academic Motivation

**Nitza Davidovitch \* and Ruth Dorot**

Department of Education, Ariel University, Ariel 40700, Israel
\* Correspondence: d.nitza@ariel.ac.il

**Abstract:** The current study is a case study examining a student association's endeavors to foster a social–academic climate on campus, grant students a sense of self-efficacy, and even contribute to students' motivation. The research literature lacks empirical knowledge on the activity of student associations and their contribution to institutions of higher education and their students. Moreover, academic institutions see student associations at times as a burden, a type of labor union to be placated by the faculty and the academic institution. The research sample consisted of 122 students from a university in Israel (38 men and 84 women; mean age 25). Several questionnaires were administered to the respondents: a questionnaire on the social–academic climate on the academic campus, a questionnaire on students' academic motivation, a questionnaire on students' self-efficacy, and a sociodemographic questionnaire. From the research findings, it is evident that the activities of the student associations on the academic campus play a meaningful role in fostering students' academic motivation and self-efficacy. The research findings indicate that the scope of student association activities is positively related to the students' academic motivation and self-efficacy. Moreover, students living in student dorms were found to evaluate the activities of the student association as higher than did students who were not living in student dorms. These findings constitute a preliminary foundation for future studies on the importance of student associations in academic institutions and their contribution to the students. Furthermore, these findings highlight the need to boost and increase student associations' activities to provide students with adapted and efficient solutions to their challenges. The student association can be transformed from a burden to an asset for the academic institution.

**Keywords:** social–academic climate; students' association; self-efficacy; academic motivation

## 1. Introduction

This study addresses the relationship between a student association on an academic campus and students' perceived social–academic climate on campus, self-efficacy, and motivation.

### 1.1. Social–Academic Climate

Classroom climate is defined as students' physical, intellectual, emotional, and social study environment, and it is determined by the overall interactions between the following factors: the mutual relations between the educational staff and the students and among the students themselves, stereotypes, demography, and cultural and social differences [1]. Classroom climate is described as a significant contributor to advancing students' achievements, facilitating their psychological well-being, and helping them become integrated and function optimally. As a result of this significant contribution, an important and major role can be ascribed to classroom climate in the personal development of individuals in an educational institution [2].

In contrast to classroom climate that includes interactions between a variety of factors, social–academic climate emphasizes the social component that relates specifically to mutual relations within the academic expanse [3]. The nature of mutual relations in an academic educational setting (student–teacher and student–student) and the interpersonal interactions between these factors have many social–emotional effects on students. Mutual relations between students and teachers are particularly significant, as teachers are students' social agents. Studies have found that warm and caring student–teacher relations based on trust and support are associated with students' emotional regulation abilities, compliance, and effective coping with future difficulties and challenges.

Regarding mutual relations among the students, positive interactions among students have been found to produce a sense of well-being, support, greater social involvement, and cooperation. Since the value that students ascribe to interactions with their peers is particularly central, these serve as important social factors for the student as well [4].

Notwithstanding the considerable importance of the teaching practice and the appropriate professional use of a variety of academic materials, the social domain is an inseparable part of the climate at the educational institution, and it has also been found to be associated with active learning and success. Indeed, it has been found that when students' social–emotional skills are enhanced, academic achievements rise, and learning reaches higher standards [5]. This finding can be further corroborated by the claim that people process information based on their experiences in relationships and social connections. As stated, feelings, social connections, and relationships are outcomes of interactions in the academic expanse. Positive relationships encourage the mind to absorb and preserve information. In contrast, negative relationships reduce the mind's ability to process information, learn, and reach high academic achievements [4,5]. This statement is supported by a meta-analysis of 270 studies by [6] found that social–emotional factors of students in a wide age range had the greatest and most significant impact on learning and achievements, more than teaching methods in class and other factors explored.

*1.2. Motivation*

The term motivation comes from the word "motion." Motivation theories and psychologists explain that motivation implies goal-oriented human behavior by an individual or group, which involves arousing, directing, regulating, or stopping a certain behavior [7]. The most well-known researchers of motivation are those of Reference [8]. Their research led to the development of the theory of self-determination, which was subsequently expanded and revised. Deci and Ryan defined motivation as an internal force that urges one to perform an action that has positive and rewarding results for the individual. This approach distinguishes between different types of motivation, all of which depend on investing energy, finding a direction, persevering in a task, and adhering to the achievement of goals, which cause an organism to act. The most fundamental distinction is between intrinsic motivation, which involves doing something because it is interesting or enjoyable, and extrinsic motivation, which involves doing something because it generates some distinct outcome. Another term is amotivation—the lack of drive, intention, perceived desire, or readiness to make an effort, given external rewards [9].

Studies found that intrinsic motivation was reduced when the target action was accompanied by tangible external rewards and prizes [10]. In addition, studies have proven that threats, orders, forcible goals, and other coercive factors reduce intrinsic motivation. The reason for this is that humans are born inquisitive and action-oriented, so they must investigate and choose independently. These actions help raise intrinsic motivation because they allow people a greater sense of autonomy [11].

Studies show that people motivated by intrinsic motivation and those motivated by extrinsic motivation differ: the intrinsically motivated have higher interest and self-confidence, perseverance, creativity, vitality, self-confidence, and general psychological well-being [12]. Other researchers agree with Deci and Ryan and say that intrinsically

interesting actions are important and that focusing on task features and their potential to arouse intrinsic interest is efficient [13,14].

## 2. The Association between Motivation and Learning

In recent decades, there has been increasing recognition of the central role of motivational processes in students' academic success and other adaptive processes, such as attitudes toward studying and inappropriate classroom behaviors. Hence, research programs aiming to understand and investigate the motivations and processes underlying the behavior of school children were developed [15]. It can be said that motivation has an important and central role in learning processes, and therefore motivation and learning are interconnected and affect each other. Researchers of motivation in schools and in various educational settings focused on students' involvement in school issues and student interest in various activities. They studied the degree of students' adherence to study goals, and their effort and perseverance, with emphasis on students' personal beliefs, such as values, goals, and how they should be achieved [9].

From a psychological perspective as well, significant weight is ascribed to motivation, and most psychologists contend that motivation is essential for efficient learning [7]. Motivation to learn comprises several elements, including students' inclination, energy, and urge to learn while performing a variety of tasks that require high abilities and considerable efficacy [10]. Motivation to learn is associated with the different aspects of the learning process, such as students' interest in learning, the pleasure they derive from their studies, the extent and level of active participation in class, and the academic achievements they attain. Therefore, a drop in a student's participation and achievements is also related to their motivation, which justifies an investigation into what has changed [10]. Another role of motivation is to change or direct human behavior, as learning reflects the behavioral change that occurs due to various psychological factors. Successful execution of tasks usually derives from the individual's motivation to do so [9]. Hence, motivation is one of the main conditions for successful learning.

As stated, the research literature divides motivators into internal factors and external factors, where the internal factors are formed within the individual, such as the desire to succeed and adherence to the goal, while the external factors exist in the individual's environment and provide extrinsic motivation, such as an anticipated reward for optimal performance of a task [9]. Some researchers claim that intrinsic motivation has a greater impact on the learner's level of motivation than extrinsic motivation, which is affected by the student's environment. Researchers found support for this, showing that when motivation comes from a positive place of enjoyment and internal desire, the outcome will be better than in the case of extrinsic motivation, when the person performing the action will display less interest [16]. At the same time, there is no doubt that extrinsic motivation too has an important part in motivating students [17]. Accordingly, it has been suggested that the educational staff be more focused on mixed motivations in their teaching practice and combine different techniques that affect both extrinsic and intrinsic motivation [7].

## 3. Motivation to Learn in a Social–Academic Sphere

Academic institutions are primary socialization agents in the lives of their students, and constitute the site of social interaction with a variety of people, leading to students' exposure to many significant academic, social, and emotional experiences [2]. Consequently, the institution is capable of giving students an equal opportunity to engage in authentic tasks that have significance outside academia, thus raising students' level of motivation [18]. Expanding students' learning skills and learning horizons is a major goal of institutions of higher education that aim to train professionals with high levels of reasoning and autonomy. The method of teaching that produces optimal learning is one that focuses on developing and nurturing students' self-value. This type of teaching focuses on structuring learning by means of an educational environment that stresses students' needs and interests, which facilitates the learning experience and their achievements [14].

The term motivation focuses on the individual's characteristics and includes the importance of social influences on learning and motivation. For example, learning in the classroom does not occur within oneself but rather while forming social ties and mutual relations between the teachers and the students, which considerably affects how students learn and their motivation to learn [19]. Many studies agree that traditional teaching methods have been found to repress students' motivation to learn. Therefore, it is important to diversify teaching methods capable of motivating students by changing the learning environment, which has also been found to contribute to students' motivation to learn [7,17,20].

The theory of self-determination stresses the individual's tendency to develop his or her potential for self-realization through the satisfaction of three basic universal needs: the need for contact and belonging, the need for competence, and the need for autonomy. Satisfying these three needs promotes optimal development, intrinsic motivation, effort, and social and emotional adaptation [21,22]. The degree to which students perceive the academic task as capable of satisfying their needs affects the quality of their motivation and fosters strong involvement, while repressing needs is detrimental to the quality of motivation and reduces its intensity. In recent years, there has been increasing criticism by researchers regarding the benefits of traditional teaching, which detracts from students' motivation. Researchers contend that teachers' use of more advanced teaching methods is very important for motivating learning; thus, the use of advanced technologies in learning constitutes an important element in promoting students' motivation to learn [9].

## 4. Self-Efficacy

In 1977, Albert Bandura introduced the social cognitive theory and the theory of self-efficacy, where he defined self-efficacy as the belief in the human capacity to generate change by enlisting motivation, cognitive resources, and courses of action necessary to respond to task requirements [23,24]. Beyond this definition, social self-efficacy can be manifested in social-cognitive self-efficacy, which involves understanding social rules and others' feelings in social situations, and in social-behavioral self-efficacy, which involves performance of expected behaviors in social situations [25]. Perceived self-efficacy is a key motivator of self-determination in learning, execution, and achievements and stimulates action, and because it is a subjective assessment of one's capabilities, it is called "self-efficacy." Self-efficacy reflects people's belief in their personal efficacy, where these beliefs are formed on the basis of life stories and experiences [26]. Several factors affect one's general outlook on life, including self-concept and self-esteem [27,28]. Hence, it is claimed that people with high efficacy beliefs function in life better than people with low efficacy [23].

Self-efficacy may be manifested in a specific area of one's life and can vary within the same area and across areas of performance [24]. Although self-efficacy can be evident in a range of areas, there are two main spheres: academic self-efficacy and social self-efficacy [29]. Academic self-efficacy refers to the belief that one can perform well in academic tasks while investing the necessary efforts. In order to execute a task optimally, it is not sufficient to acquire knowledge—one must also believe in one's ability to operate efficiently in order to achieve the goal and cope effectively with the challenges and complexities involved [2]. There are many different academic tasks, such as achieving course goals, meeting course requirements, and earning a self-satisfying grade [30]. In a study that examined the contribution of students' self-efficacy and self-esteem to their academic achievements, students with high academic self-efficacy and appreciation of their learning abilities had a higher probability of attaining better achievements than students with low self-efficacy [31].

## 5. The Association between Self-Efficacy and Motivation

As stated, self-efficacy is the belief in one's ability to perform a task, where these beliefs are formed by thoughts and feelings that come and go and that are experienced before performing a task, while performing a task, and after task completion. During this time span, a process of evaluation, judgment, and control is formed, ultimately creating

an assessment of how one operates; hence, efficacy contributes significantly to human motivation to attain achievements [32]. Moreover, self-efficacy is considered particularly central to setting goals and achieving them. It is positively correlated with effort and performance, such that people with high self-efficacy will persevere in their efforts until their goal is achieved, while those with low self-efficacy are inclined to give up easily and even avoid action [15,33].

According to social cognitive theory, people are more inclined to perform tasks that they believe they are capable of performing and become less involved in tasks that they feel they are less capable of performing. People's beliefs regarding their ability in a certain area affect their choices, their efforts, their persistence, and their resistance to obstacles or failures [27]. People create perceptions of their efficacy while processing information received from four sources [23]:

1.  Opportunities to succeed, past achievements, and previous experiences (enactive attainments): This is the most prevalent source, with the greatest influence on the process of adapting self-efficacy beliefs. This source is based on authentic experiences of control. Successes raise the assessment of one's efficacy, while failures reduce it. Every new appreciation of one's self-efficacy becomes integrated with the previous ones and further establishes it.
2.  Observing the performance of others or social persuasion (vicarious experiences): The second strongest source of information for self-efficacy consists of experiencing learning while watching a role model, or learning by emulating. Observing the successful experiences of another can raise the observer's efficacy assessment and vice versa; observing a role model with similar abilities as those of the observer and witnessing failure despite the efforts made can diminish the observer's self-efficacy assessment.
3.  Verbal persuasion: Another source of information that affects the processing of one's perceived self-efficacy is realistic persuasion by other people that the learner is capable of performing a task successfully. The power of persuasion depends on the trustworthiness, knowledge, and proficiency of the persuader. When statements such as "I trust you" are said by a person whom the learner highly admires, they affect the learner's self-efficacy evaluation, but such persuasion can also generate the opposite result when one is persuaded that one has no ability.
4.  Physiological and affective states: Sweating, palpitations, and anxiety are other sources of information for assessing one's self-efficacy. In some situations, pressure and tension are perceived as indicators of a fear of failure, inability, or lack of proficiency. Researchers who examined the four sources of efficacy corroborated Bandura's hypothesis [34] that posited that past performance and experiences of control are the sources with the greatest impact on self-efficacy beliefs.

An association was found between self-efficacy, worker motivation, and job performance [27]. In this study, the researchers assessed the effect of self-efficacy on people's job performance and the mechanism through which one's self-efficacy determines job performance and motivation. They found that it is necessary to identify the practical implications of improving workers' self-efficacy in order to motivate them and improve their performance. Hence, self-efficacy has an important and significant role in our lives and can affect one's behavior. In addition, it has been proven that self-efficacy is associated with self-control, resilience in the face of failure, efforts to perform and carry out tasks, and efficient problem solving. High self-efficacy affects one's motivation and overall self-confidence [27].

A study on the connection between students' self-efficacy in academic institutions and their personality traits, motivations for applying to the institution, their perceptions of the institution's social–academic climate, and their self-efficacy [35,36] showed that, across academic institutions and departments, in a more supportive and personal social–academic climate, students' self-efficacy increases. This study applies a multidimensional approach to teaching practice that is dependent on academic social–academic climate in an effort to

study the interactive connection between teaching practice, as perceived by undergraduate students through their sense of success, self-efficacy, and institutional social–academic climate. Teaching practice, as perceived by students, is a potentially significant factor in students' academic success. Additional studies followed this study, which focuses on the critical role of a social–academic climate as a mediating factor in higher education (e.g., [3,37–41]).

In these studies [39–41], researchers studied social–academic climate perceptions of students in various departments and academic institutions. The findings of these studies, which were conducted over a period of more than five years, demonstrate the significance that students attribute to various measures of social–academic climate and the role of these elements in students' self-efficacy and sense of success. Additional studies [42,43] found the social field constitutes an integral part of the academic climate in academic institutions and has been found to have great weight in learning success.

## 6. The National Union of Israeli Students

The National Union of Israeli Students (NUIS) is the representative organization of 350,000 academic students in Israel and the home of student associations and communities that operate throughout Israel's campuses, colleges, and universities (NUIS website). The NUIS operates through regular and thorough public and parliamentary work with national decision makers and promotes the status, power, and opportunities of students in Israel on a daily basis. The NUIS sees its role as empowering the status of students and granting three major opportunities: (a) equal opportunity for academic studies for all students, irrespective of their financial state and social or geographical affiliation; (b) opportunity to access quality education and experience a meaningful study period that is relevant for post-degree life, meeting the increasing need to impart tools and skills suitable for the changing employment world; and (c) opportunity to influence the future of Israel and to participate in social and civil involvement through a variety of programs, scholarships, and action groups implemented.

The NUIS operates in several areas:

Policy: The NUIS represents and promotes students' interests as a natural partner in decision-making centers in government offices with regard to forming policies on education. In addition, it provides assistance, support, and professional counseling to the local student associations in academic institutions and supports student campaigns on various policy issues. The NUIS policy department is responsible for four major policy areas: academic policy, technological education, employment and promotion of under-represented populations, and social infrastructure.

Community involvement: This department is responsible for the NUIS's social–civil policy, with the goal of leading social change among young people in Israel and promoting accountability and impact on the present and future of society and of the country. This department operates change communities and influence groups that allow students to take part in significant community work.

Scholarships: This department creates opportunities for students to take part in meaningful social activity, while fostering their involvement in society as active citizens and simultaneously relieving the financial burden of academic studies. The department is responsible for screening and recruiting program heads, producing conferences and seminars, training scholarship recipients, and also working regularly with various government offices and private organizations.

Developing communities and projects: This department strengthens local student associations through the ongoing support of their chairpersons, creates training courses for the professional advancement of members of the local student associations, and establishes projects that benefit students.

Research: This department supports efforts to change policy and legislation that improve the situation of young people and students in Israel by creating a body of knowledge

that is used to promote deliberation and research, and provides access to knowledge to all students in Israel.

Government and external relations: This department promotes the NUIS's agenda with decision makers in the government and Knesset. These efforts include active participation in relevant Knesset committees to promote the interests of students in Israel. In addition, the department takes action to develop and strengthen the relations between students in Israel and students around the world, creating a shared dialogue.

Publicity: This department is responsible for strengthening the NUIS's public image in the eyes of decision makers, policymakers, and the media in Israel, as a leading and reality-changing organization in the world of young people and students. The department is responsible for developing and implementing a communications strategy, and publicizing NUIS activity among the student associations and the general public.

Many times, especially in democratic regimes, student unions also engage in national political issues that are not directly related to students. In such cases, students become a social organization that is active at the national level, through various activities such as demonstrations or actions to influence governments and parliaments (e.g., student protests in France in May 1968 and student protests in the US against the Vietnam War). Across the world, the power of student demonstrations to change historical government decisions is evident, such as in the 1960s and 1970s, when students in the US and Europe demonstrated against the governments of their countries. Student unions occasionally serve as fruitful grounds for the development of young politicians, who, after gaining experience in student union politics, move to other arenas of political action.

Many studies have been conducted on the associations between student unions and political action, yet the main purpose of student unions remains to improve the conditions of students. This study directly examines the connection between a student association's endeavors to foster a social–academic climate on campus, and students' evolving sense of self-efficacy and motivation. This study offers a unique perspective based on research in the field of the perceived social–academic climate and the academic learning environment on campus, and recognizes the significant value of a social–academic climate in teaching and learning processes in all educational settings. This issue has, however, been largely neglected by institutions of higher education [44], where research is valued over teaching or community service [45]. As a result, teaching practice gives little attention to a social–academic climate in academic settings, although a historical study of the development of academic institutions in Israel highlights the changes that have occurred since the country's independence in 1948. These changes are reflected in the increased number and diversity of academic institutions as well as an increasing number of students.

## 7. Student Associations around the World—A Comparative View with Israel

Student associations, also called student unions, are organizations established by students in an academic institution to improve students' rights, well-being, and benefits and to collectively protect their shared interests and goals. Most student associations are established and operated by a group of students who study at the target institution. The heads of the association are elected in democratic elections by all students who are members of the association. In many student associations, the members, who comprise the majority of the students at the academic institution, pay an annual membership fee, and the association's budget allocations are determined in the association's budget meetings.

In many cases, student associations in distinct academic institutions form wider organizations such as regional or national student unions and associations. This political mechanism often allows the associations to work toward goals shared by all students in several academic institutions in the same area or country by creating a larger organization that has more power than the local organizations. In 1946, the International Union of Students was established in Prague, encompassing 155 organizations from 112 countries with a total of 25 million students. After some 60 years of activity, it ceased its operations.

Since then, there have been initiatives to establish a new international organization that has yet to bear fruit.

The main sources of student associations' resources are as follows:

Welfare fees: In many academic institutions, students pay welfare fees in addition to tuition. These payments are collected by the institution, and part of the money is transferred by the management to the student association as the provider of some of the welfare services provided at the institution. The amounts transferred are usually established in a written agreement between the institution administration and the association, where the association gives its commitment to provide certain services in return for the money received. Payment of the welfare fees is voluntary, but the large majority of the students choose to pay them in order to enjoy the many welfare services provided by the institution and the association.

Membership fees: Some student associations choose to collect annual membership fees from their student members instead, or in addition to the funds received as welfare fees. Most associations, however, prefer to refrain from collecting membership fees and to make do with the welfare fees, both due to the complexity of the process involved and the concern that this would significantly reduce the number of students interested in being members of the association.

Funds received from subsidiary companies: Since the associations are non-profit organizations, they are not permitted to perform economic or commercial activities and translate their reputation and the financial power of their members into financial terms. Hence, many associations establish subsidiary companies through which financial activities are performed. A considerable part of the subsidiaries' profits is transferred to the associations and used to realize the associations' goals.

As stated, the student associations operate in a wide range of areas relevant to students' lives. The main services provided by the associations to their members are as follows: (a) resolving academic problems, from the level of individual courses to that of the institution in general; (b) representing students in the disciplinary court of the academic institution; (c) maintenance of an exam bank, where students can receive exams from previous years; (d) production of textbooks, support courses, and subsidized marathons in preparation for exams; (e) granting scholarships to members of the association, on socioeconomic criteria and community service; (f) producing cultural and entertainment events on campus, such as a fair for the beginning of the school year, student's day, exhibitions and performances, the association's newspaper; (g) legal counsel and tax counseling provided to students at no cost; and (h) personal accident insurance.

Nearly all academic institutions in Israel have a student association. The associations operate as non-profit organizations. The main goals of the associations are usually identical to those of student associations around the world, namely representing student affairs versus the management of the academic institution, initiating activities for the well-being and benefit of the institution's students, and taking care of issues related to students' lives in the academic, social, financial, and public sphere.

The leaders and heads of the associations are elected by the students who are members of the associations in annual or biannual elections. Due to the associations' size, the students usually elect an assembly of representatives comprised of representatives of departments, faculties, or campuses, and the elected assembly subsequently gathers and elects the association's chairperson and the other members of its committee. Those running for positions in the association are usually members of student groups but sometimes also run independently.

In years when the government decided to raise tuition or reneged on promises made to the students, the student associations took steps to interrupt studies in most large institutions of higher education in Israel and even organized demonstrations, which became violent, and confrontations with the police in order to draw public attention to the matter. Examples are the campaign to reduce tuition in 1998 and the campaign in 2007 against the establishment of the Shohat Commission.

Nevertheless, in recent years, criticism has been leveled at the students for joining battles only when they involve their own tuition and not being involved in national matters, as are students in other countries, such as in the United States and Western Europe in the 1960s and 1970s, when students protested against the regime and changed historical government decisions.

The first official recognition of the student associations and their status was by the report of the Meltz Commission in 1996. The Commission, which addressed tuition at institutions of higher education, determined that additional fees such as welfare fees and guard duties would be voluntary and could be collected only based on an agreement between the academic institution's administration and its student association.

The Student's Rights Law, enacted in 2007, was the Israeli legislator's first reference to student associations. Clause 1 of the law defines the student association: a body elected by the students as determined by law. The law obligates the academic institution to allow elections for the student association (clause 20). The student association must operate according to regulations it sets and publicizes for all students at the institution (clause 21). Moreover, it was determined that an appeals committee for appealing decisions of the disciplinary committee would be comprised of representatives of the faculty and of the student association (clause 18).

Quite a few politicians and public figures, mayors, members of the Knesset, and government ministers began their public activity in student associations. These include Roni Milo, Limor Livnat, Silvan Shalom, Michael Eitan, Tzachi Hanegbi, Haim Ramon, Michael Reiser, Yona Yahav, Moshe Amirav, Eitan Kabel, Michael Kleiner, Gila Gamliel, Alex Miller, Boaz Toporovsky, Yair Revivo, Itzik Shmuli, and Ram Shefa.

In the past, the student associations were organized under two umbrella organizations: the National Union of Israeli Students, which included mainly the student associations of the universities and large colleges, and the Israel Student Organization, which included the student associations of the small colleges. In 2009, the Student Organization was merged with the Union of Israeli Students and ceased to operate as a separate entity. Today, NUIS is the umbrella organization of 64 student associations in Israel and represents approximately 300,000 students.

## 8. Case Study: A Student Association's Contribution to the Academic Campus

The Student Association at Ariel University was established in 2004 and is registered as a non-profit organization based on the Law of Non-Profit Organizations. It was established with the aim of providing a response to the rights and needs of students, irrespective of faith, sex, and race, while maintaining full transparency and professionalism. At the beginning of each school year, the representatives and chairperson of the Association are elected democratically by the students in all faculties, and the elected representatives of the Association elect its officials for the next year.

The Association has five main divisions. The culture division includes parties, lectures, and fairs held on campus. The academic division includes the development of workbooks for practicing study material and academic marathons for each faculty. The welfare division includes student dorms, scholarships, and employment. Spokesmanship includes publicity for events on campus, and sports include extracurricular activities and competitions.

According to the chairperson of the Student Association, the role of the Association became even more meaningful and central for students during the COVID-19 pandemic, since many students needed prompt responses from all the Association's divisions. The Association was required to ensure that adequate academic conduct was maintained while also making certain that the students' rights, both basic rights and academic rights, were upheld during the crisis so that their achievements would be maintained.

## 9. Research Purpose

The study examines whether and to what degree there is an association between the activities of the Student Association designed to foster the social–academic climate



on campus and the academic motivation and self-efficacy of undergraduate students at Ariel University.

**10. Research Hypotheses**

Upon reviewing the Association's website, the following can be hypothesized:

A positive association is found between the activities of the Student Association designed to foster the social–academic climate on campus and undergraduate students' academic motivation, such that the greater the activity of the Student Association, the higher the academic motivation.

A positive association is found between the activities of the Student Association designed to foster the social–academic climate on campus and undergraduate students' self-efficacy, such that the greater the activity of the Student Association, the higher the self-efficacy.

Differences are found between students living in the dorms and students living elsewhere in their assessments of the social–academic climate (and the activities of the Student Association).

**11. Method**

*11.1. Respondents*

Participants were 122 undergraduate students at Ariel University. The age range was 20–30, the mean age was 25.07, and the standard deviation was 2.54 (M = 25.07 and SD = 2.54). Most of the research participants were women (68.9%), and 31.1% were men. Most of the participants were single (69.7%), and 30.3% were married. Most of the participants lived in central Israel (71.3%), and the rest lived in the north (20.5%) and south (8.2%). Most of the participants worked while studying for their degree (50% held a part-time job, and 18% held a full-time job), while 32% did not work while studying for their degree. Approximately one-quarter of the participants (24.6%) lived in the student dorms, and the remainder did not live in the student dorms (75.4%). Sociodemographic characteristics of the research participants are presented in Table 1.

**Table 1.** Sociodemographic characteristics of the respondents (N = 122).

| Variable | Frequency | % |
|---|---|---|
| Gender | | |
| Men | 38 | 31.1 |
| Women | 84 | 68.9 |
| Marital status | | |
| Single | 85 | 69.7 |
| Married | 37 | 30.3 |
| Place of residence | | |
| North | 25 | 20.5 |
| Center | 87 | 71.3 |
| South | 10 | 8.2 |
| Employment | | |
| None | 39 | 32.0 |
| Part-time job | 61 | 50.0 |
| Full-time job | 22 | 18.0 |
| Living in the dorms | | |
| Yes | 30 | 24.6 |
| No | 92 | 75.4 |

Notably, our study did not include graduate students because most graduate students are less involved in the Association's activities than undergraduate students.

*11.2. Research Instruments*

The study employed several questionnaires:

A demographic questionnaire that included the students' personal background: sex, age, place of residence (north/center/south), personal status (single, married, and widowed), employment (none, full-time job, and part-time job), and residence in the student dorms (yes/no).

Questionnaire on social–academic climate. Social–academic climate is measured by a questionnaire on the Student Association's activity developed by the researchers. Participants rated their agreement with 10 items describing the Student Association's activities on a Likert-type scale from 1 (*not at all*) to 5 (*very much*). The reliability of the questionnaire in the study is $\alpha = 0.88$.

Questionnaire on academic motivation [46]. The first explained variable, academic motivation, was measured by the Motivated Strategies for Learning Questionnaire (MSLQ), which was developed by [46]. It originally comprised two parts: academic motivation and learning strategies. In the current study, the researchers used the part on academic motivation. This questionnaire utilizes a Likert scale and includes 31 items representing six scales of academic motivation: intrinsic motivation, extrinsic motivation, task value, personal beliefs regarding control of learning, expectations for success and self-efficacy (which constitute a joint scale), and test anxiety. Participants rated their agreement with each item on a scale from 1 (*not true at all*) to 7 (*very true*). A higher score implies higher motivation. The reliability of the questionnaire in our study is $\alpha = 0.93$.

Questionnaire on self-efficacy [47]. The second explained variable, self-efficacy, was measured by a questionnaire that originally related to teachers' perceived efficacy. It was translated into Hebrew [48], and the items were adapted to students' perceived efficacy. This questionnaire utilizes a Likert scale and includes 15 items. Participants rated their agreement with each item on a scale from 1 (*strongly agree*) to 6 (*strongly disagree*). A higher score implies higher self-efficacy. The reliability of the questionnaire in our study is $\alpha = 0.83$.

*11.3. Research Procedure*

The questionnaires were administered online via Google Docs and distributed through the WhatsApp application to all courses and disciplines at Ariel University, and included undergraduate students, defined as the research population. The survey packet began with an explanation to participants that their responses would be used for an academic study at Ariel University and that the study is intended for undergraduate students at Ariel University only. It was also explained that participants must complete the questionnaire independently and that their personal information and responses would be anonymized and be used only for the purpose of the study. Finally, we provided them with contact details to be used in case of any questions or problems that might arise while completing the questionnaire.

## 12. Data Analysis

The data were analyzed statistically with SPSS software version 26.0. For the purpose of a sociodemographic description of the sample, frequency distributions and percentages were calculated for categorical variables, and means and standard deviations were calculated for continuous variables. To test the first and second research hypotheses, Pearson correlations were utilized. To test the third research hypothesis, we conducted a t-test for independent samples. Cronbach's alpha reliability was used to examine the reliability of all the research variables.

## 13. Research Findings

The current study is a case study examining a student association's endeavors to foster a social–academic climate on campus, grant students a sense of self-efficacy, and even contribute to students' sense of motivation. We shall now present the research findings.

### 13.1. Descriptive Statistics

The means and standard deviations of the research variables are shown in Table 2. It is evident that academic motivation has the highest mean of the three research variables (M = 5.69), followed by self-efficacy with a high mean (M = 4.52), and finally, social–academic climate, with a mean of M = 3.60.

**Table 2.** Means and standard deviations of the research variables.

| Variable | N | M | SD |
|---|---|---|---|
| Academic motivation | 122 | 5.69 | 0.71 |
| Self-efficacy | 122 | 4.52 | 0.71 |
| Activity of the Student Association for creating a social–academic climate | 122 | 3.60 | 0.67 |

### 13.2. Internal Reliability

The internal reliability measures of the research variables are shown in Table 3. The variable of academic motivation includes some 31 items with internal reliability of 0.93. The variables of self-efficacy included 15 items with internal reliability of 0.83 (of these, items 4, 9, and 15 are reversed), and the variable of social–academic climate included 10 items with internal reliability of 0.88. Reliability is considered adequate if it is higher than 0.7; therefore, the three variables have good reliability.

**Table 3.** Values of the internal reliability measure (Cronbach's $\alpha$).

| Variable | Item | Scale | Reverse Items | Alpha Value |
|---|---|---|---|---|
| Academic motivation | 31-1 | 7-1 | | 0.93 |
| Self-efficacy | 15-1 | 6-1 | 4, 9, 15 | 0.83 |
| Social–academic climate (activity of the Student Association) | 10-1 | 5-1 | | 0.88 |

### 13.3. Exploring the Hypotheses

Research hypothesis 1: A positive association is found between the activities of the Student Association designed to foster a social–academic climate on campus and the level of academic motivation of undergraduate students, such that the greater the activity of the Student Association, the higher the academic motivation. For this purpose, we conducted a Pearson correlation.

Table 4 indicates that, consistent with the hypothesis, a positive association was found between the Student Association's activities and academic motivation ($r_p(122) = 0.66$, $p < 0.01$), such that the greater the activities of the Student Association, the higher the academic motivation. The hypothesis was confirmed.

**Table 4.** Pearson correlations between the research variables.

| Variables | | | |
|---|---|---|---|
| | 1 | 2 | 3 |
| 1. Academic motivation | - | | |
| 2. Self-efficacy | 0.66 ** | - | |
| 3. Social–academic climate (activities of the Student Association) | 0.53 ** | 0.37 ** | - |

** $p < 0.01$.

Research hypothesis 2: A positive association is found between the activities of the Student Association designed to foster a social–academic climate on campus and the level of self-efficacy among undergraduate students. For this purpose, we conducted a Pearson correlation.

Table 4 indicates that, consistent with the hypothesis, a positive association was found between the Student Association's activities and the level of self-efficacy ($r_p(122) = 0.37$, $p < 0.01$). The greater the activity of the Student Association, the higher the self-efficacy. The hypothesis was confirmed.

Research hypothesis 3: Differences will be found between the assessments of the social–academic climate on campus between students living in the dorms and students living elsewhere. For this purpose, we conducted a t-test for independent samples. Consistent with the research hypothesis, a difference was indeed found between the groups. The assessment of the Student Association's activities was significantly higher among students living in the dorms (M = 3.90 and SD = 0.44) than among students living elsewhere (M = 3.51 and SD = 0.70) ($t = 2.83_{(120)}$, $p < 0.01$), supporting the hypothesis.

## 14. Conclusions and Discussion

Like their international counterparts, student associations in Israel were established for the well-being and benefit of students and to protect students' shared interests and goals. The first official recognition of a student union and its status was in 1996, in the context of a campaign regarding tuition hikes in public academic institutions in Israel. In 2007, the Students' Rights Law was passed, marking the first legislative act directed at student unions. In Israel, 64 student associations are members of the national student association (NSUI). These associations span all types of academic institutions and are active on diverse issues that are relevant to the lives of students, such as the production of cultural and entertainment events on campus, art exhibitions, and the publication of newsletters. Student associations have focused on creating a positive social–academic climate on their respective campuses.

This study explored the association between the activities of student associations designed to foster a social–academic climate on campus and students' self-efficacy and academic motivation, through a case study of the activities of a student association at one university in Israel. The contribution of the study is in being a pioneer study on this topic in Israel. Findings indicate a consistent trend regarding the association between the activities of the Student Association in fostering a social–academic climate on campus and students' self-efficacy and motivation. The research findings show that considerable meaning and importance can be attached to the activity of the Association and its enhancement at institutions of higher education, as this variable was found to contribute to the students' sense of self-efficacy and academic motivation.

The findings of this study point to the challenge facing academic institutions with respect to their attitude toward student associations and interactions with student association activists. Support by academic institutions may serve to incentivize and leverage the activities designed to create a significant social–academic climate for students in such a significant period in students' lives. This is true globally, and in the reality of Israel, where students typically begin their academic studies at a later age, after completing two or three years of military service, this is an especially important aim.

At Ariel University, the administration recognizes the importance of its student association. Following the University's Articles of Association, the chair of the association is regularly invited to administration and Senate meetings. Student representatives are members of Senate committees, including the disciplinary and appeals committees. The Dean of Students maintains regular contact with the Student Association. In addition to these ongoing interactions, the institution also supports the association, for example, by providing financial support for Students' Day, on which the institution cancels all classes.

The institution collaborates with the Student Association in all the informal and cultural activities that it organizes for the students to enhance social solidarity, develop students' aesthetic sense and moral reasoning, and create an optimal social and academic climate on campus. These events offer students valuable experiences during their studies, and many times are events in which students would otherwise not be able to participate. The Student Association creates opportunities for students to become involved in social

events as well as activities that impact others, in which students "practice" how to be constructive citizens.

Thus, the student association has the potential to serve as a valuable partner in the civil–social education of students and in students' empowerment, far beyond its activities in the political sphere. The findings of this study indicate the importance of institutional support to a student association, and an understanding that the student association functions as the institution's "right hand" by attending to students' needs and concerns, protecting their rights, and creating a social–cultural–academic climate on campus.

Therefore, academic institutions are advised to view student associations through this lens rather than focus on associations' militant and political actions. Together, academic administrations and student associations can work together to reinforce the social–academic climate on campus, empower students, and give them a valuable opportunity to practice their skills as productive citizens in society.

Further studies might examine the contribution of student associations' activities at other academic institutions, including colleges, both public and private, also distinguishing between student associations' activities in student dorms and those with no activity in the dorms, or by student characteristics such as age, personal status, and other factors.

There is room to examine the association between involvement in student association activities and leadership ability, the role of the student association in developing students' social skills, and the mutual relations between faculty and students who are members of the association. These research questions make it possible to explore the topic under investigation from many perspectives and to contribute to the expanding knowledge on the contribution of student associations' activities to not only the students, but also to the academic institution in general.

**Author Contributions:** N.D. and R.D. contributed equally to conceptualization, methodology, validation, formal analysis, investigation, resources, data collection, and writing. All authors have read and agreed to the published version of the manuscript.

**Funding:** This research received no external funding.

**Institutional Review Board Statement:** Not applicable.

**Informed Consent Statement:** Not applicable.

**Data Availability Statement:** Data sharing is not applicable.

**Conflicts of Interest:** The authors declare no conflict of interest.

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
