# Peer review of "On the Relationship between a Student Association’s Endeavors to Foster a Social–Academic Climate on Campus, Students’ Self-Efficacy, and Academic Motivation"

_education, doi:10.3390/educsci13070647_

Round 1

Reviewer 1 Report

In general, the study is well justified and elaborated, so it can be an interesting contribution. I found this to be a socially and scientifically relevant paper. The article has exactly clear aim. The authors give a review of contemporary research in the field. The article is well structured and clearly written, including introduction, objectives of the study, data and methods, analysis and results. However, I recommend expanding the discussion by also adding implications for practice, as well as indications for the development of future research. 

Finally, another important issue is the references. Reviewers are kindly requested to review the most recent references. 

The language needs to be revised. Authors need to edit the paper with the help of a native speaker. 

Author Response

The paper was edited by a professional English-language editor. 

We expanded the discussion and added practical implementations, as requested. 

We noted several directions for future research, as requested.

We updated the reference list and added more recent references, as requested. 

Reviewer 2 Report

 This an interesting study though I am not sure that it is a "pioneer study". A student association wishes take measures such as ..... to foster a social-academic climate on campus, which they speculate will support a sense of self-efficacy as well as contributing to student motivation in academic matters. Self-efficacy has been extensively researched over the last 20-30 years though in my experience it has been connected to the development of academic achievement and self-worth. In simple terms it is about the ability to do something and the belief and confidence to do actually do it. Collective efficacy is important in a range of situations including interprofessional work. If professionals vouch for each other and believe in their combined ability to address a variety of issues, each needing professional knowledge from different directions. 

This author has done a thorough job of providing an overview of the research literature, tough it almost too detailed. The style is rather heavy and almost cumbersome given the three topics being addressed. I think it might be useful to be slightly unconventional is this paper. I suggest that it is reorganised into three mini-papers, each with its own literature preceding the statistical results  and finishing with a short comment on the findings. Then to finish the paper there is a comparative section drawing attention to the different approaches and commenting on the relationship between them. 

I would also like to see some more speculative discussion on the connections between the three areas, and the extent to which they are complement each other. 

Finally it might be of interest to add a few interviews to the data background or alternatively say a bit more about the strengths and weaknesses for the particular purpose for which the surveys are used in this rather unusual research. 

Were three different studies carried out?

What can be learnt about gender issues?

What difficulties are particular to the geographical and political context?

see above

Author Response

The paper was edited by an English-language editor. 

This research is a pioneer study that examines the relationship between the student union’s activities and social-academic climate and students’ self-efficacy. It was conducted as a single study (and not three separate studies), and therefore dividing it into three separate papers would defeat its purpose. The theoretical foundation for these relationships are in the literature review and also explored in previous studies of the author and her colleagues.   We strengthened the theoretical foundation and also elaborated in the discussion section.

We expanded the discussion section to include our thoughts on the connections between the variables of interest. 

At this stage, we did find it appropriate to conduct additional interviews, although this suggestion will be certainly be applied in our next research project on this topic.  The strength of this study lies in its highlighting of the contribution of the student association to the academic setting and campus, especially today, within a world of digital tools and AI. A strong relationship between institutional administrative and the student union and its activities is especially important and the study shows that this relationship potentially has a strong impact on students and their success.

We leave the issue of gender effects to a follow-up study. 

Finally, we found no special difficulties or challenges that are particular to the geographic and political context of our study (Israel), especially because our study focused on the student association's social role and not its political role. 

Reviewer 3 Report

Thank you for providing me with the opportunity to review this manuscript. I appreciate the authors’ effort to investigate the association between the student associations’ activities and students’ self-efficacy and motivation in the context of Israel, where student unions play a significant role in the country’s education and political landscape. Within this context, the authors examined the correlations between students’ self-efficacy and motivation and their perception of the activities of their student union in an Israeli university.

 Although these might have implications for Israel's higher education, this study has a number of significant limitations that preclude me from recommending its acceptance. Two primary concerns are as follows:

 1.       At the conceptual level, the paper fails to compile relevant literature to organize a valid argument for the hypotheses. First, it remains unclear how social-academic climate is defined theoretically and why it could be operationalized as the activities of the Student Association. Second, In the section “Motivation to learn in a social-academic sphere”, the authors mainly talked about how teaching style/approaches affect motivation, which is not relevant to the paper’s topic. Third, while they seemingly articulated the research on motivation, the authors did not cite the relevant literature properly.  For example, it is noticeable that the majority of the research cited in this paper was conducted among children or school students, yet the authors did not make it clear why these findings would hold for university students. In addition, the authors tend to use “Studies show…” or “Many studies agree…” or “In recent years there is increasing criticism…” without any references.

2.       Empirically/methodologically, the correlation analysis is weak and insufficient to support the authors’ claim that “…students association’s endeavors to foster a social-academic climate on campus, grant students a sense of self-efficacy…”. 

The quality of the English language seems fine to me.

Author Response

The paper was edited by a professional English-language editor. 

With respect to the conceptual level, we expanded in the paper on the issues in the reviewer's comments. We stress that the strength of this study is its exploratory nature. To the best of our knowledge, this is the first study in the era of growing digitization of academic studies, that explores a student association's contribution to the social-academic climate on campus and the relationship between social-academic climate and important aspects of students' success, specifically self-efficacy and academic motivation. We hope that future studies will generate additional insights on these relationships in this and other contexts, and explore additional facets of the impact of student associations on students' success. Such insights can help academic institutions leverage student associations to promote students' success. 

Round 2

Reviewer 3 Report

Thank you for your further explanation and revision, which has not addressed my concerns sufficiently.

The quality of English Language is fine.

Author Response

Dear Reviewer, 

We made additional corrections to the paper according to the Editor's comments. Thank you for your helpful comments.